# Two-Photon Fluorescence in Red and Violet Conjugated Polymer Microspheres

**Yanyan Zhi [1], Ziwei Feng [1,2], Tanisha Mehreen [3], Xiaoyuan Liu [3], Kirsty Gardner [3], Xiangping Li [1], Bai-Ou Guan [1], Lijuan Zhang [3], Sergey I. Vagin [4], Bernhard Rieger [4]** and **Alkiviathes Meldrum [3,\***

[1] Guangdong Provincial Key Laboratory of Optical Fiber Sensing and Communications, Institute of Photonics Technology, Jinan University, Guangzhou 510632, China; yanyanzhi@jnu.edu.cn (Y.Z.); fzwyunlinji@163.com (Z.F.); xiangpingli@jnu.edu.cn (X.L.); tguanbo@jnu.edu.cn (B.-O.G.)

[2] Wuhan National Laboratory for Optoelectronics (WNLO), Huazhong University of Science and Technology (HUST), Wuhan 430074, China

[3] Department of Physics, University of Alberta, Edmonton, AB T6G 2E1, Canada; tmehreen@ualberta.ca (T.M.); xiaoyuan@ualbertaa.ca (X.L.); kgardner@ualberta.ca (K.G.); lijuan@ualberta.ca (L.Z.)

[4] Wacker Chair of Macromolecular Chemistry, Technical University of Munich, Lichtenbergstraße 4, 85747 Garching bei München, Germany; vagin@tum.de (S.I.V.); rieger@tum.de (B.R.)

\* Correspondence: ameldrum@ualberta.ca

**Abstract:** We investigate the two-photon fluorescence (TPF) of conjugated polymer (CP) microspheres with diameters up to tens of micrometers. Two polymers, emitting in either the violet or red, were first synthesized and characterized in terms of their one-photon fluorescence and three-dimensional internal microstructure. Under femtosecond infrared excitation, both types of microspheres showed a strong TPF, which was investigated by the excitation intensity dependence, emission spectroscopy, time-resolved luminescence, and photobleaching dynamics. While the violet-fluorescent microspheres performed similarly compared to dye-doped polystyrene counterparts emitting at a similar wavelength, the red-fluorescent microspheres showed a two-orders-of-magnitude stronger TPF. This excellent performance is attributed to enhanced hyperpolarizability associated with intermolecular interactions in the polymer solid, indicating a route toward designed CP microspheres that could outperform currently-available microparticles for sensing or imaging applications involving two-photon fluorescence.

**Keywords:** conjugated polymer; microspheres; fluorescence; two-photon fluorescence

## 1. Introduction

Two-photon fluorescence (TPF) has been widely applied in fluorescence microscopy for biological imaging and sensing applications [1,2]. Compared to one-photon fluorescence (OPF) imaging, TPF methods reduce the background noise arising from the excitation of material away from the laser focus [3]. Near-IR excitation additionally reduces the possibility of photo-damage and is especially beneficial for biological imaging, due to the transparency window of most tissues in the IR spectral range [4]. In particular, dye-doped polymer microspheres are used for two-photon fluorescence imaging and diagnostic sensing applications [5]. Fluorescent microspheres with diameters of 10 μm embedded in turbid media show better image quality in two-photon fluorescence due to the smaller scattering cross-section of the illumination beam and reduced out-of-focus fluorescence [6]. TPF of microspheres can also be used for calibration of laser trapping systems [7] and for up-conversion lasing with a low threshold [8]. TPF of dye-doped microspheres has been used for temperature and refractive index sensing [9,10], ultra-sensitive bio-affinity assays [11,12], endoscopy [13], and optical tweezing [14]. Moreover, the quantitative sensing of DNA has been achieved by measuring the TPF emission intensities of hybrid-quantum-dot/polystyrene microspheres [15]. TPF of dye-doped polymer microspheres also shows considerable potential in cancer theranostics [16].

The applications of multi-photon fluorescence can be hindered by the low absorption cross sections of common dyes [17,18] and by the limited dye concentrations that can be required to minimize self-quenching effects in conventional polymer microspheres. Conjugated polymer (CP) microspheres could potentially provide an alternative platform for TPF microscopy, biological imaging, lasing, and sensing applications due to their combined electronic, optical, and mechanical properties [19–25], potentially large optical two-photon absorption cross sections [26,27], and low cytotoxicity [28]. The polyphenylene-vinylene polymers (PPVs), including bis-alkoxy-substituted derivatives such as poly[2-methoxy-5-(2-ethylhexyloxy)-1,4-phenylene vinylene] (MEH-PPV), possess relatively high two-photon absorption coefficients and have a one-photon fluorescence quantum efficiency typically ranging from 5–40% depending on their conformation and state (e.g., dissolved in a specific solvent vs. in the solid form [29–32]). TPF has been reported in an MEH-PPV/polystyrene blended waveguides [33], but there have so far been only sparse examples of CP microspheres for TPF applications [34] and none for the common, widely-used, and relatively easily-fabricated polyphenylene vinylenes or poly-p-phenylenes.

Following recent discoveries in the synthesis of small CP microspheres [34–37] and much larger MEH-PPV microspheres [38], we aimed to investigate the TPF properties of CP microspheres with widely different "colors" and compare them to those from conventional dye-doped fluorescent microspheres showing TPF. To do this, we synthesized a relatively low molecular weight MEH-PPV (red emission) and a violet-emitting counterpart (poly[2-methoxy-5-(2-ethylhexyloxy)-p-phenylene]; MEH-PPP) that are both soluble in organic solvents. We use these precursors to produce large (up to ~100 μm) fluorescent microspheres via a solvent emulsification method. We then examine the internal structure of the particles, demonstrate and quantify the two-photon fluorescence for both emission colors, evaluate the one-photon and two-photon fluorescence characteristics, speculate on the origin of the TPF, and compare the magnitude of the two-photon response against that of conventional dye-doped polystyrene microspheres.

## 2. Experimental Methods

### 2.1. Polymer Synthesis

MEH-PPP (Figure 1a) was synthesized following variations on methods reported in [39–42] in accordance with the Grignard metathesis polymerization method (GRIM), in which we used a 1,4-dibromo-2-(2-ethylhexyloxy)-5-methoxybenzene pre-monomer. The latter was obtained via established methods [39]. The pre-monomer was subsequently dissolved in dry degassed tetrahydrofuran (THF) along with 5.4 mL of 1 M turbo-Grignard reagent ($^i$PrMgCl · LiCl) in THF. The result was then stirred for nearly one day at room temperature. Next, 2.3 mg (0.05%) of [NiBr$_2$(PPh$_3$)(I$^i$Pr)] catalyst (synthesized according to previous methods [40]) was dissolved in 0.2 mL THF and added to the reaction solution. After a full five days the reaction mixture became an opaque mass. This was mixed into methanol with a few drops of concentrated HCl added and rapidly stirred for several minutes. A precipitate settled to the bottom of the beaker, after which it was decanted with methanol. The suspension was mixed again for several hours and decanted once again. It was then stirred over one night, the methanol was decanted, and finally the result was washed several times with acetone. After the final drying in-vacuo, the obtained yield was typically on the order of 1 g. The resulting MEH-PPP was characterized by NMR spectroscopy (Figure S1) and by gel permeation chromatography (GPC) (Figure S2), yielding molecular weights of $M_w$ = 150 kDa and $M_n$ = 30 kDa, with some variations in different batches.

MEH-PPV (Figure 1b) with molecular weights of $M_w$ = 140 kDa and $M_n$ = 30 kDa, as measured by gel permeation chromatography, was synthesized and characterized according to previous methods [38]. The polymer materials were stored in the dark and in a rough vacuum of ~100 mTorr before the microsphere fabrication. The NMR and GPC spectra for MEH-PPV are shown in Figure S3 and Figure S4, respectively.

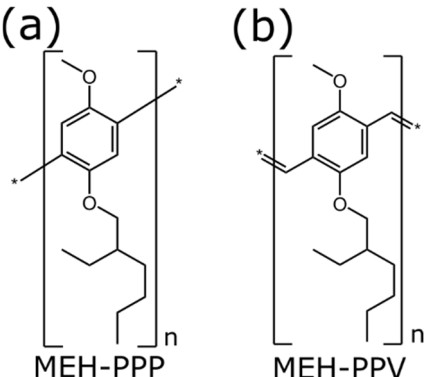

**Figure 1.** Molecular structures of the repeat units (**a**) MEH-PPP and (**b**) MEH-PPV.

### 2.2. Microsphere Fabrication

The microspheres synthesized from MEH-PPV (red fluorescence) or MEH-PPP (violet fluorescence) were fabricated using a solvent emulsification method following [38]. In brief, 0.5 mL of 1-propanol (99.5%, HPLC Grade, Sigma-Aldrich, Burlington, MA, USA) was added to a 10 mL solution of deionized (DI) water with dissolved poly(vinyl alcohol) (PVA, $M_w \approx 13{,}000–23{,}000$ Da, 98% hydrolyzed, Sigma-Aldrich, 50 g L$^{-1}$). The MEH-PPV or MEH-PPP was dissolved in chloroform at a concentration of 20 g L$^{-1}$ and was then injected via a syringe through a length of thin polytetrafluoroethylene tubing whose opening was submerged in the water solution. The injection flow rate was 0.5 mL/min for one minute (0.5 mL total), while the solution was rapidly stirred with a magnetic stir bar (Figure S5). The solution was then stirred for an additional 4 h to evaporate the chloroform, leaving solid polymer microspheres that gradually settled to the bottom of the beaker. The microspheres were then washed with DI water several times in order to remove the PVA. For OPF and TPF investigations, the microspheres were dropped onto a glass slide using a pipette. The water then evaporated, leaving behind a large number of "dry" microspheres which could then be chosen for laser excitation and two-photon fluorescence. In order to reduce the photo-oxidation, as clearly shown for MEH-PPV in [43], the fabricated microspheres were stored in milliQ water in a dark environment prior to the fluorescence measurements. At all times we tried to minimize photo-oxidative degradation of the microspheres by storing them in the dark and away from ambient air.

### 2.3. Optical Measurements

For OPF imaging, the microspheres were placed on the stage of a Nikon TE2000e (Melville, NY, USA) inverted epifluorescence microscope and excited with a 365-nm or 405-nm LED. Fluorescence spectroscopy was performed by pumping an ensemble of microspheres using the combined 352 nm and 364 nm lines of an Ar ion laser. The fluorescence was collected with an optical fiber, sent through a 375- or 400-nm long-pass filter, and analyzed using an intensity-calibrated miniature spectrometer. Time-resolved fluorescence was performed using a 405-nm picosecond diode laser with a single-photon counting system from Becker-Hickl (SPC-130EMN, Berlin, Germany) and an HPM-100-50 photomultiplier tube. Microstructural characterization was performed on a Zeiss Orion helium ion microscope (HIM). Three-dimensional tomographic mapping of the CP microspheres utilized a Zeiss Xradia Versa X-ray microscope (Jena, Germany) with a capillary condenser. The sample was prepared by depositing the microspheres on a thin wooden cylinder that could be easily rotated under the X-ray beam. Subsequent imaging optics (i.e., a zone plate, scintillator, and imaging CCD) were used to collect the projections. Reconstruction and image analysis were performed in the Dragonfly software package.

For the TPF experiments, the emission from individual conjugated polymer microspheres was excited using a tunable femtosecond pulsed laser (Chameleon-Ultra II at 80 MHz from Coherent, Santa Clara, CA, USA). The excitation wavelengths were 700 and 800 nm for the MEH-PPP and MEH-PPV microspheres, respectively. The polarization of

the laser light was controlled by a half waveplate and a Glan-Thompson (GT) polarizer to optimize the PL intensity, and the laser power was controlled using an attenuator in order to reduce any photodamaging of the microspheres (Figure S6). An objective lens (UPLXAPO 20X from Olympus, Tokyo, Japan, NA = 0.75, FL = 180 mm from Edmund Optics, Barrington, NJ, USA) focused the laser light onto a single microsphere. The fluorescence was collected with the same objective lens, passed through a shortpass dichroic mirror, and sent to an imaging camera (DCC3240M from Thorlabs, Newton, NJ, USA and MER-125-30UC from Daheng Imaging, Beijing, China), spectroscopy system (Shamrock 303i from Andor, Northern Ireland, UK), or single photon avalanche diode (SPAD) (PD-050-CTC-FC from Micro Photon Devices, Bolzano, Italy). The signal from the photon counting detector was then analyzed by a time-correlated single photon counting module (PicoHarp 300 from PicoQuant, Berlin, Germany). Comparison with conventional microspheres was performed by using dye-doped polystyrene microspheres whose spectral characteristics were similar to the CP microspheres synthesized in this work (specifically, "Suncoast Yellow" from Bangs Labs, Fishers, IN, USA and blue fluospheres F8837 from Thermo Fisher, Carlsbad, CA, USA).

### 3. Results and Discussion

The microspheres mainly ranged from 5–20 μm in diameter and followed lognormal size distributions in both samples (Figure 2), yielding a mean diameter and variance of 14.5 μm and 36.7 μm$^2$ for the violet particles and 12.1 μm and 23.3 μm$^2$ for the red microspheres, respectively. Any differences in the size statistics result from differences in the droplet sizes (due, for example, to possible viscosity differences), the number and size of internal voids, and uncertainties associated with experimental parameters such as the stir bar spin speed.

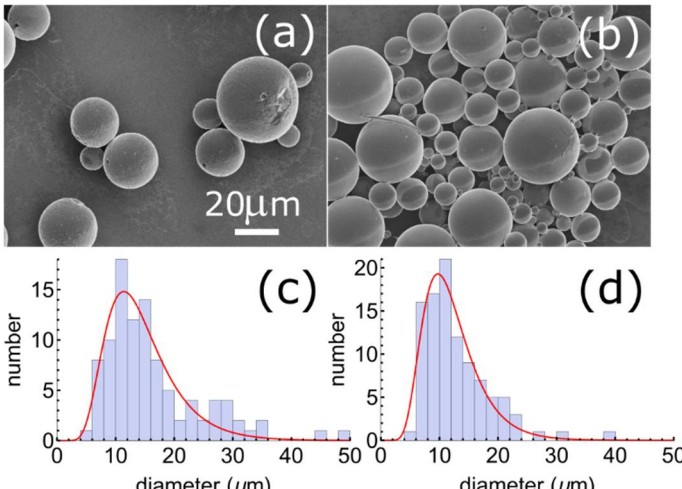

**Figure 2.** Secondary electron helium ion micrographs of MEH-PPP (**a**) and MEH-PPV (**b**) microspheres. The corresponding size distributions for MEH-PPP (**c**) and MEH-PPV (**d**) microspheres, along with a lognormal fit.

In order to further verify these ideas, a sample was fabricated using simultaneous independent injections of the violet and red polymer solutions. There was no mixing of the resulting polymers (Figure S7), showing that the size distribution is not caused by merging droplets; they must, instead, break up during the mixing cycle. The droplets enter the water phase with a uniform distribution having an approximate mean diameter on the order of 1 mm (Figure S5). Given the known solute concentration and density of MEH-PPV, one would, therefore, expect a microsphere diameter on the order of ~300 μm, which is larger than any of the observed particles. This further suggests that a "break up" rather than a merging scenario causes the observed size distributions.

The internal structure is difficult to observe by conventional methods in electron or ion microscopy (as in Figure 2a,b). Instead, we used X-ray microscopy (XRM) with tomographic reconstruction and focused ion beam (FIB) sectioning to report the internal structure of both types of CP microspheres. A 3D XRM shadow view of several MEH-PPP (violet) microspheres is shown in Figure 3a, two separate planar sections separated by 1.5 μm are shown in Figure 3b,c, and Figure 3d presents a FIB cut through a single MEH-PPP microsphere imaged with a helium ion beam. Finally, Figure 3e–h presents the corresponding set of imaging data for MEH-PPV (red) microspheres. The MEH-PPP microspheres often had one or two dimples intersecting the surface, whereas the MEH-PPV microspheres showed only much smaller scattered internal voids. These differences were more-or-less consistent throughout all the observed microspheres, despite the identical nature of the synthesis methods, the general similarity of the polymer repeat units (Figure 1), and the similar molecular weights.

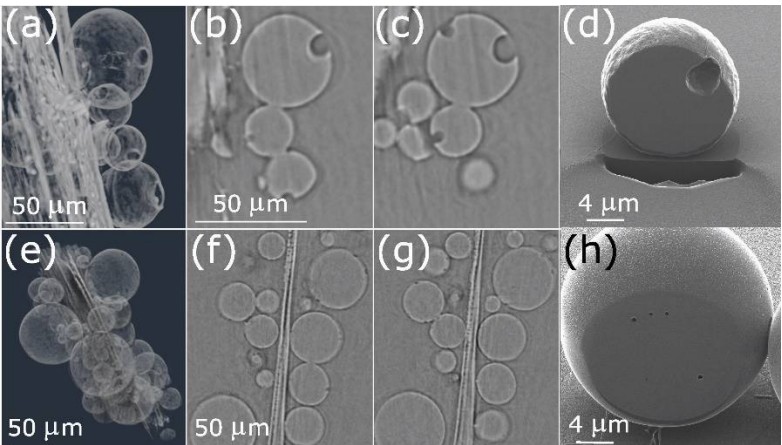

**Figure 3.** X-ray microscopy tomographic reconstructions of MEH-PPP in 3D view (**a**) and in 2D slices separated by 1.5 μm in vertical distance (**b,c**). An FIB cut through a representative single violet microsphere is shown in (**d**). The MEH-PPP microspheres generally contain single large dimples near the surface. (**e–h**) A similar sequence of tomographic reconstructions and FIB cut for MEH-PPV (see online Visualization 1). A few voids or holes are present but are smaller and more uniformly scattered throughout the particle.

Although the internal 3D reconstruction of CP microspheres has not been investigated before, conventional polymers systematic studies have shown that the interior structure is dependent on the degree of intermixing of the polymer solvent and of the continuous phase, as well as the polymer concentration and molecular weight [44]. Since the solubility of water in chloroform is only ~0.05%, the incorporation of a non-solvent droplet is unlikely to be responsible for the relatively large void structures in the MEH-PPP microspheres. Instead, the voids could mark the locations where the chloroform concentrated prior to the solidification of the polymer particle. These observations suggest that the entrapped solvent droplets are able to reorganize more easily in the process of forming MEH-PPP particles, ultimately coagulating into a large near-surface droplet.

### 3.1. One-Photon Absorption and Fluorescence

The microsphere fluorescence spectra show that the MEH-PPP and MEH-PPV particles fluoresce in the violet (ca. 393 nm) and red (ca. 640 nm), respectively, with a Stokes shift from the absorption maxima obtained from solid films of the same polymers, which peaked at 320 nm and 500 nm, respectively (Figure 4). The MEH-PPV microspheres have a broad PL spectrum with strong evidence of interchain interactions that lead to an extended red tail up to ~750 nm, as reported in solid MEH-PPV films [45]. The absolute quantum efficiency was 9% for solid MEH-PPV and 60% for solid MEH-PPP, respectively. The microspheres were readily observable by one-photon fluorescence microscopy (Figure 4a,b), although in

this imaging mode the internal defects found by XRM and focused ion beam sectioning were difficult or impossible to observe.

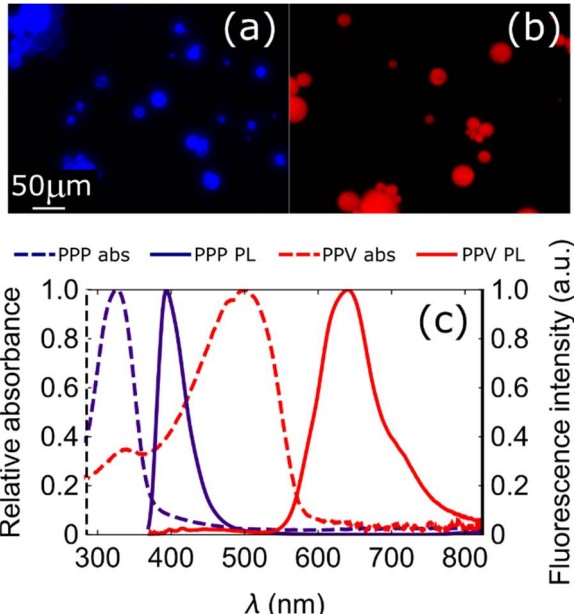

**Figure 4.** Fluorescence images of MEH-PPP (**a**) and MEH-PPV (**b**) microspheres. (**c**) Absorption and emission spectra of PPV and PPP thin films, and microsphere ensembles, respectively.

### 3.2. Two-Photon Fluorescence

Individual violet microspheres showed clear evidence of two-photon fluorescence when pumped at 700 nm (140 fs pulses) with the laser spot focused on the center of a single microsphere via an objective lens (Figure 5a). The emission spectrum maximized near 400 nm, although it was cut off in the UV due to the microscope optical system. Thus, the violet emission spectrum was preserved in the TPF case, but there was a long-wavelength tail that was not present in the OPF spectrum, which could be due to interchain interactions similar to the extended red tail observed in other CPs [45]. The small peaks around 412 and 435 nm in the single-sphere TPF spectra likely result from the optical system (see the Supporting Information and Figure S8 for a more detailed discussion of system-induced spectral artifacts).

The violet-fluorescent microspheres photobleached when exposed to an irradiance of 2 mW (~210 kW/cm$^2$; Figure 5b). Photobleaching under TPF excitation conditions is commonly observed and its rate is dependent on the excitation irradiance [27,46]. The decrease in the integrated intensity fit well to a double exponential decay model with time constants of 8.25 ± 0.04 min and 0.70 ± 0.01 min, respectively, yielding a weighted mean bleaching time of 2.22 ± 0.02 min. When using a pump power of 40 μW, however, there was only limited photobleaching over this timeframe. The emission intensity ($I$) vs. pump power ($P$) curve was generated at low power in order to minimize photobleaching and the resulting data were fit to a power law, $I = AP^n$, where $A$ is just a scaling parameter and the fit exponent $n = 2.02 ± 0.02$ (Figure 5c), in good agreement with the well-known TPF power law mechanism.

The one- and two-photon time-resolved photoluminescence (TRPL) of the violet microspheres yielded non-single-exponential decay dynamics (Figure 6a,b). Care was taken to minimize photobleaching by using a much lower power in the TRPL experiments (~50 μW in this case). For both situations (OPF and TPF) the decays were fit with a numerical model that convolved the measured instrument response function with a lifetime distribution (Equation (1)).

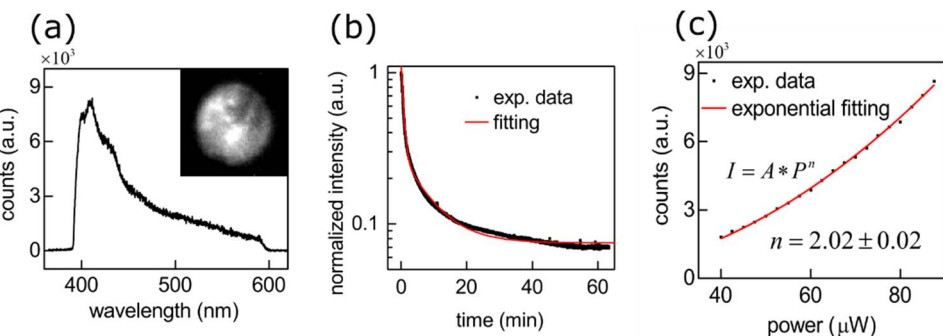

**Figure 5.** Fluorescence data from MEH-PPP microspheres. (**a**) Two-photon fluorescence spectrum from a single microsphere. The inset shows an TPF image captured by a monochrome camera (diameter of the microsphere is ~15.0 μm). (**b**) The normalized fluorescence intensity of a single microsphere as a function of time at an excitation power of 2 mW. The red curve represents a double exponential fitting using Equation (S2). (**c**) Two-photon fluorescence intensity as a function of the incident laser power. The red curve shows a power law fit which yielded an exponent of 2.02 ± 0.02.

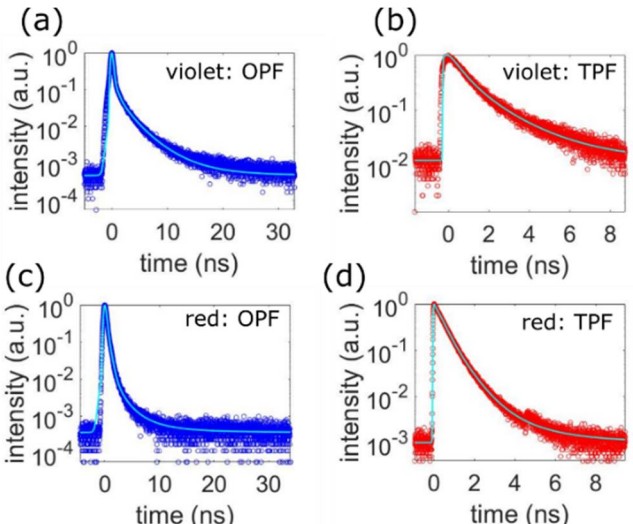

**Figure 6.** Fluorescence dynamics and model fits (Equation (1)) for OPF (blue points) and TPF (red points). (**a**) MEH-PPP one-photon dynamics (**b**) MEH-PPP two-photon dynamics (**c**,**d**) the same as above but for MEH-PPV.

Several models were attempted, including a double exponential, a lognormal, a Gaussian, a Gamma distribution, and weighted combinations of these with a single exponential. In both cases (OPF and TPF), a good fit (as determined by the sum of the absolute values of the residuals) was found for the linear combination of a fast single exponential and a lognormal distribution, given by the following model:

$$I = \left( B \cdot e^{-t/\tau_0} + (1-B) \int_0^\infty H(\tau) \cdot e^{-t/\tau} d\tau + c \right) * (IRF) \qquad (1)$$

where $H(\tau)$ represents a lognormal distribution of decay time constants given by

$$H(\tau) = \frac{1}{\Delta\tau_1} \frac{e^{-\frac{(\mathrm{Ln}(\tau)-\tau_1)^2}{2(\Delta\tau_1)^2}}}{\sqrt{2\pi\tau}} \qquad (2)$$

In these equations, $B$ determines the relative strength of the single exponential and lognormal contributions, $\tau_0$ is the single exponential time constant, $c$ is the offset, and $\tau_1$ and $(\Delta\tau_1)^2$ are the median and variance of the lognormal distribution, respectively. Notably,

the OPF was dominated by the single exponential component whereas for the TPF the lognormal decay component was much stronger (see Table S1 for the fit parameters). In other words, the lognormal part of the decay function was significantly enhanced in the two-photon fluorescence dynamics.

Finally, in order to establish a benchmark, we compared the TPF intensity of the MEH-PPP microspheres with commercial F8837 blue fluorescent microspheres with similar diameters. While the dye used is unfortunately unknown (manufacturer claims a commercial secret), we found that under identical conditions of ~210 kW/cm$^2$ irradiance, the commercial microspheres were about twice as bright as the MEH-PPP microspheres (Figure S9a). When corrected for absorption differences at half the excitation wavelength, the emitted intensities were about the same for the two types of microspheres in both the OPF and TPF cases. Thus, in contrast to the case for MEH-PPV microspheres discussed below, the violet CP microspheres perform roughly similar to conventional dye-doped polystyrene microspheres in terms of their TPF emission.

The MEH-PPV microspheres showed a strong red TPF when excited at 800 nm with the pulsed laser spot focused on the center of a single microsphere (Figure 7a). A short-pass filter was used to block the excitation laser light, resulting in the observed cut-off wavelength around 682 nm in the TPF spectrum. The broad and asymmetric TPF spectrum is consistent with the extended red emission from interchain interactions in thick MEH-PPV films [47]. In some microspheres, there were smaller oscillations with an approximate mean spacing of ~4.8 nm (Figure S9), which likely arise from the spherical whispering gallery modes. The MEH-PPV microspheres also photobleached under intense IR excitation corresponding to an irradiance of approximately 2 mW (~150 kW/cm$^2$). The photobleaching likewise followed a double exponential model quite well (Figure 7b), yielding a mean decay time of 6.22 ± 0.07 min. The fluorescence did not recover after one-hour laser "off time" (Figure S10), and encapsulating a microsphere into a resin did not significantly reduce the effect. Thus, bleaching is likely related to a non-reversible photochemical process that may not depend on exposure to atmospheric oxygen, possibly similar to case reported for organic dyes [48]. The emission intensity was nearly quadratic with the excitation power density (taken at low power to minimize bleaching), again closely fitting a power law, with an exponent of 1.96 ± 0.01 (Figure 7c). The TPF emission spectrum was approximately 1700 times stronger than was the case for MEH-PPP microspheres when the average excitation power was 40 μW (Figure S11).

The OPF and TPF fluorescence dynamics of the red microspheres also fit well to a single exponential combined with a lognormal lifetime distribution (Figure 6c,d; Table S1). Most notably, as with the violet particles, the OPF has a stronger single exponential component while the lognormal part is much stronger in the TPF. Previous work on the MEH-PPV TRPL from drop cast and spin coated films was fit using a double exponential decay [49], where the longer lifetime component was attributed to the effect of aggregates. Considering that there is likely a distribution of aggregated states and possible inter- (and intra-) molecular interactions in solid MEH-PPV, a distribution of lifetimes rather than two discrete ones is reasonable. Thus, at least two pieces of evidence point to interchain effects as playing a significant role in the microsphere TPF: the presence of long-wavelength features in the emission spectra and the greater contribution of the lognormal lifetime distribution component in the two-photon TRPL, which is consistent with a distribution of emissive states. Thus, it is likely that the increased dipole strength associated with interchain electronic interactions is responsible for the strong TPF performance of these microspheres. This deduction is in agreement with a recent investigation of conjugated polyelectrolytes, where the two-photon fluorescence intensity increased as a function of the concentration of interchain-coupled aggregates [50].

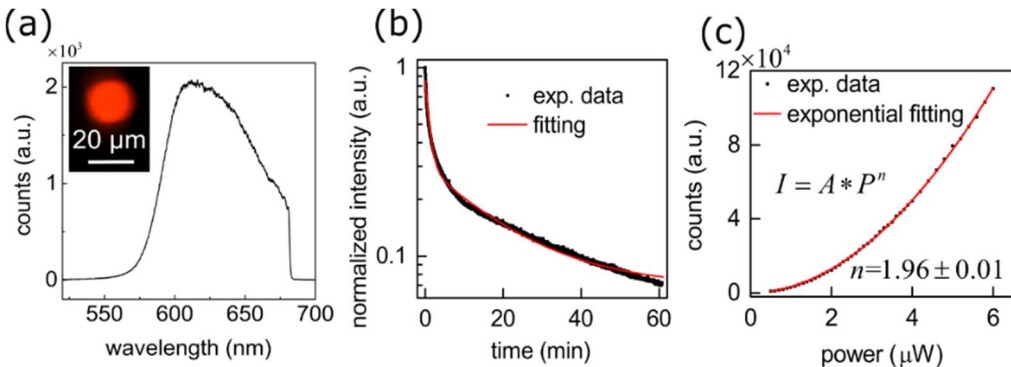

**Figure 7.** Fluorescence data from MEH-PPV microspheres. (**a**) Two-photon fluorescence spectrum from a single microsphere. The inset shows an epifluorescence image captured by a color camera. (**b**) The normalized fluorescence intensity of a single microsphere as a function of time at an excitation power of 2 mW. The red curve represents a double exponential fitting. (**c**) Two-photon fluorescence intensity as a function of the incident laser power. The red curve shows a power law fit which yielded an exponent of $1.96 \pm 0.01$.

The two-photon absorption cross section for MEH-PPV is known to be ca. 1000 GM for similar excitation wavelengths to those used here [26]. Given the much lower TPF intensity for the violet-emitting MEH-PPP, the large inherent absorption cross section of the red microspheres must vastly overwhelm their lower quantum efficiency in comparison to violet-emitting ones. The TPF efficiency of conjugated polymers is related to the conjugation length and the degree of aggregation [51], where, as previously discussed, aggregation can either create new pathways for two-photon fluorescence or quench it by lowering the emission quantum efficiency. The hyperpolarizability of CPs is specifically affected by the presence or absence of a donor-$\pi$-acceptor motif, the density of $\pi$ electrons, the conjugation length [52], and intermolecular interactions in the solid state [53]. While it is beyond the scope of this work to untangle the electronic origins of the TPF precisely, the structural similarity of the two polymers' repeat units and sidechains suggests that the latter two reasons are likely related to the excellent performance of the red-emitting microspheres. Finally, we note that both the solvent and the casting conditions play key roles in the solid polymer conformation and the degree of interchain effects [54,55]. Contrary to the case for OPF, aggregation may play a beneficial role for TPF; thus, for a given CP one may expect improved TPF performance by choosing a solvent that leads to more open molecular conformations in the solid state.

In order to further verify the feasibility of CP microspheres for two-photon fluorescence, we finally compared the MEH-PPV microsphere TPF response to that of standard dye-doped polystyrene microspheres with similar emission spectra. Although we cannot entirely rule out the possibility of confounding photo-oxidative aging effects, care was taken to store the materials away from light and air prior to analysis (see the experimental section). For the red analogue, we selected FSSY007 "Suncoast Yellow" from Bangs Laboratories due to its similar emission spectrum as compared to MEH-PPV (again, the dye used would not be disclosed by the manufacturer). For similarly-sized microspheres, the TPF peak count rate of the MEH-PPV sphere was over 200 times larger than that of the dye-doped polystyrene microspheres (Figure S6b). The conventional microspheres showed similar bleaching time constants as compared to their CP analogues, indicating that the potentially huge increases in TPF output are, fortunately, not strongly counterbalanced by photobleaching concerns when comparing the two types of materials. Thus, we conclude that CP microspheres produced by these methods can show a significantly enhanced performance for TPF applications.

## 4. Conclusions

We fabricated structurally similar red- and violet-fluorescent conjugated polymer microspheres and investigated their 3D microstructural character and two-photon fluorescence. Using XRM and FIB methods, the 3-dimensional subsurface structure of both types of microspheres was investigated and the subsurface voids were clearly observable. A detailed analysis of the one-and two-photon fluorescence characteristics was next performed. For both polymers, the one- and two-photon emission spectra were broadly similar and both were readily observable under standard imaging conditions. Conventional dye-doped polymer microspheres are used for TPF applications currently (i.e., fluorescence imaging and diagnostic sensing applications), but they lack some of the remarkable properties of CPs, including the inherent fluorescence with high TPF cross sections. This work combines these factors to investigate the TPF properties of CP microspheres.

The fluoresce intensity, lifetime, and photobleaching effects were characterized for both types of CP microspheres. Large differences were found in the intensity of the TPF for similarly-sized red- and violet-fluorescent microspheres, despite the similarities in molecular structures of the two polymers. This difference is likely due to a significantly enhanced two-photon absorption cross section in MEH-PPV, related to a longer conjugation length and especially to interchain interactions associated with the extended red emission and corresponding lognormal tail of the decay time constant distribution. We finally compared the TPF intensity to that obtained for analogous dye-doped polystyrene microspheres from well-known commercial sources with similar linear absorption and emission spectra. While the violet microspheres showed only a similar or slightly weaker OPF and TPF emission intensity (by a factor of ~1–2), we find a much better response (by 1–2 orders of magnitude) for the red-emitting microspheres. The results indicate that CPs can potentially offer significantly better performance over that of currently-available materials.

**Supplementary Materials:** The following supporting information can be downloaded at: https://www.mdpi.com/article/10.3390/inorganics10070101/s1. Reference [56] is cited in the Supplementary Materials.

**Author Contributions:** Conceptualization, Y.Z. and A.M.; methodology, Y.Z., T.M., K.G., B.-O.G., Z.F., X.L. (Xiaoyuan Liu), L.Z., S.I.V.; software, X.L. (Xiangping Li); validation, Y.Z. formal analysis, X.L. (Xiangping Li), Y.Z.; investigation, Y.Z.; resources, A.M., B.R.; writing—original draft preparation, Y.Z., A.M.; writing—review and editing, Y.Z., A.M.; funding acquisition, A.M., B.R. All authors have read and agreed to the published version of the manuscript.

**Funding:** National Sciences and Engineering Research Council of Canada; Guangdong Province Key Field R&D Program Project (2020B0101110002); Natural Science Foundation of Guangdong Province General Program (2022A1515011475); National Natural Science Foundation of China (NSFC) (61805103); The Local Innovative and Research Teams Project of Guangdong Pearl River Talents Program (2019BT02X105); Guangzhou Science and Technology Plan Project (201904020032).

**Data Availability Statement:** Data underlying the results presented in this paper are not publicly available at this time but may be obtained from the authors upon reasonable request.

**Conflicts of Interest:** The authors declare no conflict of interest.

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
