# Peer review of "Two-Photon Fluorescence in Red and Violet Conjugated Polymer Microspheres"

_inorganics, doi:10.3390/inorganics10070101_

Round 1

Reviewer 1 Report

In this manuscript, the authors reported the synthesis of two polymer microspheres, MEH-PPP and MEH-PPV, with diameters of tens of micrometers. Due to the introduction of two polymers with the emission in either red and violet, the synthesized polymer microspheres exhibited two-photon fluorescence properties, which could be further used for biosensing and bioimaging. It is an interesting work. The experiments are good-designed and the manuscript is good-organized and written. All the conclusions are supported by the presented data. Based on these points, this manuscript is recommended for publication after minor revision.

Special comments:

1.     It is necessary for the authors to indicate clear the novelty and significance of this work compared to traditional two-photon fluorescence materials.

2.     The formation mechanism of MEH-PPP and MEH-PPV microspheres is not provided. The authors should provide more information on the self-assembly and polymerization of co-polymers for the formation of microspheres.

3.     In Figure 2, the authors utilized SEM to characterized the synthesis polymer microspheres. It is necessary to use spectral characterizations to prove the successful synthesis of both polymer microspheres.

4.     In Figure 4c, the authors presented the absorption and emission spectra of PPV and PPP thin films and microsphere ensembles. From this figure, the spectra of microsphere ensembles can not be found. In addition, why using the PPV and PPP thin films for measuring the absorption and emission?

5.     What is the potential applications of the synthesized MEH-PPP and MEH-PPV microspheres in TPF field? More discussion is needed.

Reviewer 2 Report

This article is comprehensive, logically organized, and contains valuable information on the two-photon fluorescence (TPF) in the red and violet conjugated polymers (CPs) microsphere. The authors did excellent research on investigating the structurally similar red- and violet-fluorescent CPs, namely Poly[2-methoxy-5-(2-ethylhexyloxy)-p-phenylene] (MEH-PPP), and Poly[2-methoxy-5-(2-ethylhexyloxy)-1,4-phenylene vinylene] (MEH-PPV), microspheres with diameters up to tens of micrometers and investigated their 3D microstructural character and TPF. The authors demonstrated that the red-fluorescent microspheres showed a two-orders-of-magnitude stronger TPF while the violet-fluorescent microspheres performed similarly compared to dye-doped polystyrene counterparts emitting at a similar wavelength.  It is suggested the authors should measure and discuss the findings of the 1H and 13C NMR of the MEH-PPP and MEH-PPV CPs to gain a better understanding of the chemical structure. It is also suggested the authors should place the GPC spectrogram of the MEH-PPP and MEH-PPV CPs for comparison purposes. The submitted manuscript has significant scientific insights and the conclusions are soundly supported by the experimental data. However, the present submission requires minor revisions before being considered for publication in the Special Issue: Colorimetric and Fluorescent Chemosensors for Metal Ions in the esteemed Inorganics in its current condition.

Author Response

Reviewer 2:

This article is comprehensive, logically organized, and contains valuable information on the two-photon fluorescence (TPF) in the red and violet conjugated polymers (CPs) microsphere. The authors did excellent research on investigating the structurally similar red- and violet-fluorescent CPs, namely Poly[2-methoxy-5-(2-ethylhexyloxy)-p-phenylene] (MEH-PPP), and Poly[2-methoxy-5-(2-ethylhexyloxy)-1,4-phenylene vinylene] (MEH-PPV), microspheres with diameters up to tens of micrometers and investigated their 3D microstructural character and TPF. The authors demonstrated that the red-fluorescent microspheres showed a two-orders-of-magnitude stronger TPF while the violet-fluorescent microspheres performed similarly compared to dye-doped polystyrene counterparts emitting at a similar wavelength. It is suggested the authors should measure and discuss the findings of the 1H and 13C NMR of the MEH-PPP and MEH-PPV CPs to gain a better understanding of the chemical structure. It is also suggested the authors should place the GPC spectrogram of the MEH-PPP and MEH-PPV CPs for comparison purposes. The submitted manuscript has significant scientific insights and the conclusions are soundly supported by the experimental data. However, the present submission requires minor revisions before being considered for publication in the Special Issue: Colorimetric and Fluorescent Chemosensors for Metal Ions in the esteemed Inorganics in its current condition.

Thank you very much for the comments. We have added these points to the SI. The GPC and 1HNMR spectra are now presented in Figs. S1-S4, for both polymers. A brief discussion is also provided in the SI.
